# Hyperspectral Image Super-Resolution via Adaptive Dictionary Learning and Double $\ell_1$ Constraint

**Songze Tang [1,\*], Yang Xu [2], Lili Huang [3,4] and Le Sun [5]**

1   Department of Criminal Science and Technology, Nanjing Forest Police College, Nanjing 210023, China
2   School of Computer Science and Engineering, Nanjing University of Science and Technology, Nanjing 210094, China; xuyangth90@njust.edu.cn
3   School of Science, Guangxi University of Science and Technology, Liuzhou 545006, China; lili771012@jit.edu.cn
4   School of Computer Science and Technology, Jinling Institute of Technology, Nanjing 211169, China
5   School of Computer and Software, Nanjing University of Information Science and Technology, Nanjing 210044, China; sunlecncom@nuist.edu.cn
*   Correspondence: tangsz@nfpc.edu.cn; Tel.: +86-137-7055-8030

**Abstract:** Hyperspectral image (HSI) super-resolution (SR) is an important technique for improving the spatial resolution of HSI. Recently, a method based on sparse representation improved the performance of HSI SR significantly. However, the spectral dictionary was learned under a fixed size, empirically, without considering the training data. Moreover, most of the existing methods fail to explore the relationship among the sparse coefficients. To address these crucial issues, an effective method for HSI SR is proposed in this paper. First, a spectral dictionary is learned, which can adaptively estimate a suitable size according to the input HSI without any prior information. Then, the proposed method exploits the nonlocal correlation of the sparse coefficients. Double $\ell_1$ regularized sparse representation is then introduced to achieve better reconstructions for HSI SR. Finally, a high spatial resolution HSI is generated by the obtained coefficients matrix and the learned adaptive size spectral dictionary. To evaluate the performance of the proposed method, we conduct experiments on two famous datasets. The experimental results demonstrate that it can outperform some relatively state-of-the-art methods in terms of the popular universal quality evaluation indexes.

**Keywords:** hyperspectral image super-resolution; sparse representation; adaptive dictionary learning; double $\ell_1$

## 1. Introduction

Hyperspectral sensors can capture images with many contiguous and very narrow spectral bands that span the visible, near-infrared, and mid-infrared portions of the spectrum [1,2]. Thus, hyperspectral image (HSI) can provide fine spectral feature differences, to distinguish various materials, which can be widely and successfully used for many applications, such as object classification [3,4], tracking [5], recognition [6], and remote sensing [7,8]. Due to various hardware limitations, real captured HSI usually has low spatial resolution (LR), which significantly limits its application. However, it is not effective to enhance spatial resolution by improving the imaging quality of the hyperspectral sensors, and a breakthrough in hardware will be difficult and costly. Alternatively, HSI super-resolution (SR) has been proposed to generate a high spatial resolution (HR) HSI by fusing a high spectral resolution image, such as HSI, with an image, such as panchromatic image [9–16] or a multispectral image (MSI) [17–30].

## 1.1. Related Work

Traditionally, spatial–spectral image fusion methods fuse an LR HSI with an HR panchromatic image (single band), such as pansharpening [9]. As we know, the famous pansharpening methods include intensity-hue-saturation (IHS) [10,11], high-frequency information injection [12,13], and model-based methods [14–16]. Owing to the limited spectral resolution of the panchromatic image, these methods often produce some spectral distortions. Accordingly, fusing the LR HSI with an HR MSI (see Figure 1) has attracted increasing attention. Spectral unmixing and sparse representation have become the mainstream methods for the above fusion [17,18]. Several HSI SR methods have worked by using these approaches [19–40]. In the following, we will briefly review the two categories of studies.

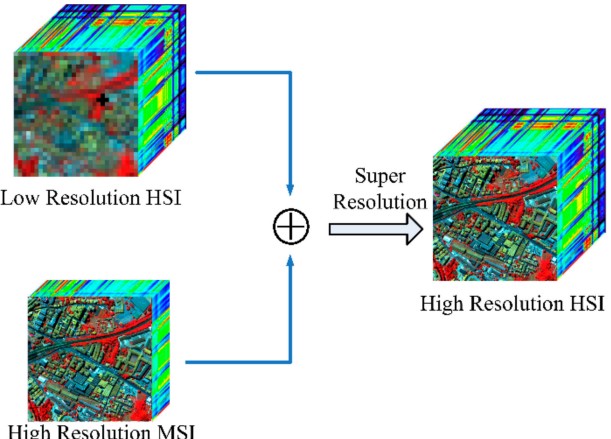

**Figure 1.** Hyperspectral image (HSI) super-resolution (SR) is generated by fusing a low-resolution (LR) HSI with a high-resolution (HR) multispectral image (MSI).

### 1.1.1. Spectral Unmixing Based Methods

In these methods, latent HSI is often decomposed into endmember and abundance matrices. The unmixing strategy [23,24] was first applied to the HSI SR problem. Naoto Yokoya et al. [19] proposed to reconstruct the HR HSI from a multispectral image and the corresponding HSI by a coupled nonnegative matrix factorization (CNMF) approach. Although this work provided a very promising result, the solution of nonnegative matrix factorization (NMF) is often not unique [20,21], and the results are not always satisfactory. By taking the sensor observation models into consideration, Bendoumi et al. [22] divided the whole image into several sub-images. Thus, the HSI could be enhanced with small spectral distortions. In general, the abovementioned methods have to unfold the three-dimensional data of the MSI or HSI into two-dimensional matrices. To break this traditional practice, the HSI is expressed by a tensor with three modes in the coupled sparse tensor factorization (CSTF) method [26]. Further, Dian et al. [27] presented a novel HSI SR approach by incorporating both the nonlocal self-similarity idea and tensor factorization into a unified framework.

### 1.1.2. Sparse Representation-Based Methods

Inspired by signal sparse decomposition, the final HR HSI is represented by an appropriate spectral dictionary learned from the real captured HSI [28–30]. A sparse regularization term was carefully designed in [28], where Qi et al proposed fusing HSI and MSI within a constrained optimization framework. Taking different spatial and spectral properties into account, the spatial and spectral fusion model (SASFM) used sparse matrix factorization to enhance the resolution of the input HSI [32]. Thus, correlations of signals in adjacent hyperspectral channels were exploited, based on the assumption that signals in different channels were jointly sparse in suitable dictionaries [33]. Recently, a nonnegative structured sparse representation (NSSR) method [38] was investigated to reconstruct an HR HSI from LR HSI and HR RGB images. To explore the global-structure and local-spectral self-similarity, a

self-similarity constrained sparse-representation (SSCSR) model was proposed by Han et. al [39]. Owing to the complex structures in HSI, superpixel-based sparse-representation (SSR) model can extract the spectral features effectively [40].

### 1.2. Motivation and Contributions

Although the aforementioned methods gave us impressive recovery performance, we can produce even better fused results from two aspects. On the one hand, the spectral basis (or the dictionary) generally has many atoms, whose sizes range from hundreds to hundreds of thousands, according to the training data. Thus, learning an adaptive size dictionary can represent the data compactly and accurately. On the other hand, due to the nonlocal correlations existing in natural images [41,42], the sparse coefficients are not randomly distributed. Therefore, it is possible to generate the high spatial resolution HSI faithfully by exploiting the nonlocal similarity of these coefficients.

In this paper, we propose an adaptive dictionary learning and double $\ell_1$ regularized sparse-representation model for HSI SR. In particular, this novel model mainly contains three parts. First, the proposed method learns an adaptive spectral dictionary whose atoms reflect the spectral signatures of materials in the HSI. It should be noted that the learning framework learns the spectral dictionary and estimates the number of atoms concurrently. Then, a transformed dictionary can be generated by choosing the corresponding bands from the adaptive spectral dictionary, which reflects the spectral signatures of the MSI. Due to the nonlocal correlations present in natural images, it is impossible for the sparse coefficients to be distributed randomly. To exploit the nonlocal similarity, a double $\ell_1$ model is used to characterize the corresponding coefficients, which are obtained by decomposing pixels on the transformed dictionary. Finally, the HR HSI is estimated by the adaptive spectral dictionary and the coefficients. The detailed flowchart is presented in Figure 2.

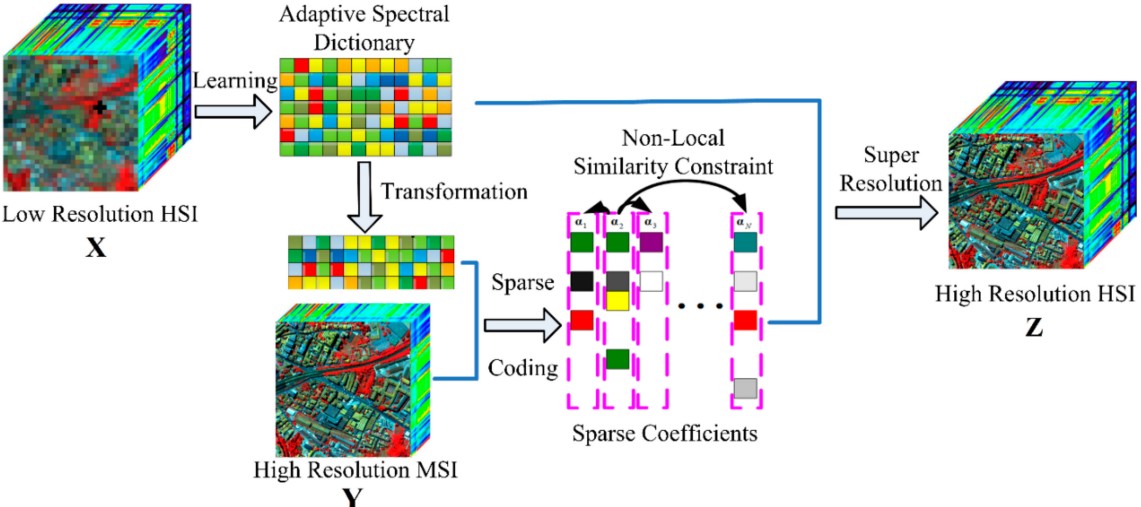

**Figure 2.** The flowchart of the proposed method.

The proposed method has the following distinct features.

(1) To represent the complex structures of HSI more effectively, an efficient adaptive size strategy is introduced to learn the spectral dictionary instead of using a fixed size dictionary.

(2) The proposed adaptive learning framework helps save time and effort in finding the correct sizes according to the content of the HSI.

(3) To improve the performance of HSI SR, a double $\ell_1$ constraint of the sparse coefficients, based on the adaptive-size dictionary, is exploited to capture the nonlocal similarity the spectral-spatial information.

(4) The proposed model can be easily and effectively solved. In addition, extensive experimental results on different HSI datasets validate the superiority of our method.

## 2. Mothodology (or Materials and Methods)

### 2.1. The Traditional Sparse-Representation Method

Let $\mathbf{X} \in R^{B \times n}$ be the LR HSI, where $n = a_1 \times a_2$. Terms $a_1$ and $a_2$ are the width and height of $\mathbf{X}$ in spatial resolution, respectively. Term $B$ indicates the spectral dimension. The acquired HR MSI is $\mathbf{Y} \in R^{b \times N}$, where $N = a_3 \times a_4$, $a_3 \gg a_1$, $a_4 \gg a_2$ and $B \gg b$. Therefore, HSI SR aims to estimate an HR HSI $\mathbf{Z} \in R^{B \times N}$ by using $\mathbf{X}$ and $\mathbf{Y}$. In general, the relationship between $\mathbf{X}$, $\mathbf{Y}$, and the latent HSI $\mathbf{Z}$ can be modelled as follows:

$$\mathbf{X} = \mathbf{ZH} \tag{1}$$

$$\mathbf{Y} = \mathbf{PZ} \tag{2}$$

where $\mathbf{H} \in R^{N \times n}$ denotes a degradation operator, and $\mathbf{P} \in R^{b \times B}$ is a transform matrix.

As a promising fusion technique, sparse representation has achieved great success [32,34,38,39]. In these methods, each pixel, $z_i \in R^B$, in $\mathbf{Z} \in R^{B \times N}$ is often represented via the following linear combination:

$$z_i = \mathbf{D}\alpha_i + e_i \tag{3}$$

where $\mathbf{D} \in R^{B \times K}$ represents a spectral dictionary, $\alpha_i \in R^K$ is the corresponding representation coefficient assumed to be sparse, and $e_i$ is the representation error.

According to Equation (1), each pixel, $x_i \in R^B$, can also be represented as follows:

$$\mathbf{x}_i = \sum_{j=1}^{n} \mathbf{H}_{i,j} z_i = \sum_{j=1}^{n} \mathbf{H}_{i,j} (\mathbf{D}\alpha_i) = \mathbf{D} \sum_{j=1}^{n} \mathbf{H}_{i,j} \alpha_i$$
$$= \mathbf{D}\beta_i + t_i \tag{4}$$

where $\beta_i \in R^K$ is the representation coefficient in the spectral dictionary, $\mathbf{D}$, $\mathbf{H}_{i,j}$ denotes the element of the $i$-th row and the $j$-th column, and $t_i$ is the residual. According to Equations (2) and (3), pixel, $y_i \in R^b$, of the MSI, $\mathbf{Y}$, can be mathematically formulated as follows:

$$\mathbf{y}_i = \mathbf{P}z_i \approx \mathbf{PD}\alpha_i \tag{5}$$

By combining Equations (4) and (5), it is easy to obtain the spectral dictionary, $\mathbf{D}$, as well as the corresponding coefficients matrix, $\mathbf{A} = [\alpha_1, \alpha_2, \dots, \alpha_N] \in R^{K \times N}$. Finally, we can generate the HR HSI, $\mathbf{Z}$, from the following equation:

$$\mathbf{Z} \approx \mathbf{DA} \tag{6}$$

### 2.2. The Proposed Method

Due to the complex structures in HSI, it is not reasonable that different variations are represented by a dictionary with a fixed size in traditional methods. In other words, the traditional sparse-representation model cannot represent the complex structures of HSI accurately. Thus, we propose the learning of a spectral dictionary with an adaptive size, according to the context in the captured area. Furthermore, double $\ell_1$ prior of the sparse coefficients is exploited to improve HSI SR quality.

#### 2.2.1. Adaptive Spectral Dictionary Learning

Traditionally, the spectral dictionary $\mathbf{D} \in R^{B \times K}$ was usually learned from a set of training exams $\{x_1, \dots, x_n\}$:

$$\min_{\mathbf{D},\mathbf{B}} \frac{1}{n} \sum_{i=1}^{n} \left\{ \|x_i - \mathbf{D}\beta_i\|_2^2 + \lambda \|\beta_i\|_1 \right\} \tag{7}$$

where $\lambda$ is a balance parameter, and $\mathbf{B} = [\boldsymbol{\beta}_1, \boldsymbol{\beta}_2, \ldots \boldsymbol{\beta}_n]$ is the corresponding coefficients matrix. $K$ is the dictionary size, which was selected empirically. As previously mentioned, a spectral dictionary with a fixed size cannot reflect complex structures of HSI accurately. Thus, we can select the dictionary size adaptively by introducing a size penalty. Motivated by [43], the size penalty was modelled with a row-sparse norm. Based on this concept, we alternatively define $\hat{\boldsymbol{\beta}}_j = \left[\beta_{1,j}, \ldots, \beta_{n,j}\right] (1 \le j \le k)$, where $\beta_{i,j}$ denotes the $j$-th element of $\boldsymbol{\beta}_i$. Then, the traditional dictionary-learning framework can be updated to the following equation:

$$\min_{\mathbf{D},\mathbf{B}} \frac{1}{n} \sum_{i=1}^{n} \left\{ \|x_i - \mathbf{D}\boldsymbol{\beta}_i\|_2^2 + \lambda \|\boldsymbol{\beta}_i\|_1 \right\} + \mu \sum_{j=1}^{K} E\left(\hat{\boldsymbol{\beta}}_j\right) \tag{8}$$

where $E\left(\hat{\boldsymbol{\beta}}_j\right) = \begin{cases} 0 \; if \; \hat{\boldsymbol{\beta}}_j = \mathbf{0} \\ 1 \; otherwise \end{cases}$, and $\mu$ is a balance parameter. For zero vectors, the size penalty outputs 0; otherwise, the result is 1. It should be noted that the dictionary is not initially constrained to containing zero vectors. Fortunately, Equation (8) can automatically determine the number of nonzeros. Thus, we can learn a spectral dictionary with an adaptive size through this model.

The objective function in Equation (8) contains multivariate indicator terms. Inspired by [44], we introduce a penalty $\mathbf{R}_\delta(\boldsymbol{b})$ to avail optimization, which is defined as follows:

$$\mathbf{R}_\delta(\boldsymbol{b}) = \min_{v \in R^n} \delta \|\boldsymbol{b} - v\|_2^2 + E(v) \tag{9}$$

If the parameter $\delta$ is large enough, $\mathbf{R}_\delta(\boldsymbol{b})$ can successfully approach the multivariate indicator function, $E\left(\hat{\boldsymbol{\beta}}_j\right)$ [44].

In summary, the adaptive spectral dictionary, $\mathbf{D}$, will be obtained through the following optimization problem:

$$\min_{\mathbf{D},\mathbf{B},\mathbf{V}} \frac{1}{n} \sum_{i=1}^{n} \left\{ \|x_i - \mathbf{D}\boldsymbol{\beta}_i\|_2^2 + \lambda \|\boldsymbol{\beta}_i\|_1 \right\} + \mu \sum_{j=1}^{K} \left[ \delta \|\hat{\boldsymbol{\beta}}_j - \hat{v}_j\|_2^2 + E\left(\hat{v}_j\right) \right] \tag{10}$$

where $v_i$ is the column vector, and $\hat{v}_j$ denotes the corresponding row vector in $\mathbf{V}$. The optimization of Equation (10) performs in an alternative scheme over three stages, which correspond to Equations (11)–(19).

In the "dictionary update" stage, the sparse coefficient $\mathbf{B}$ and the variable $\mathbf{V}$ are fixed, and we can obtain the dictionary as follows:

$$\min_{\mathbf{D}} \mathbf{L}(\mathbf{D}) = \min_{\mathbf{D}} \frac{1}{n} \sum_{i=1}^{n} \|x_i - \mathbf{D}\boldsymbol{\beta}_i\|_2^2 \tag{11}$$

The stochastic gradient descent algorithm [45] is employed to update $\mathbf{D}$ iteratively. In the *it*-th iteration, we get the following equation:

$$\mathbf{D}^{(it)} = \mathbf{D}^{(it-1)} - \delta_g \nabla_D \mathbf{L}(\mathbf{D}^{(it-1)}) \tag{12}$$

where $\delta_g$ is a learning rate, and $\nabla_{\mathbf{D}}$ represents the gradient operator of $\mathbf{D}$. We substitute $\nabla_D \mathbf{L}(\mathbf{D}^{(it-1)}) = \frac{1}{n} \sum_{i=1}^{n} \left(x_i - \mathbf{D}^{(it-1)}\boldsymbol{\beta}_i\right)\boldsymbol{\beta}_i^T$ into Equation (12) and obtain the following:

$$\mathbf{D}^{(it)} = \mathbf{D}^{(it-1)} - \delta_g \frac{1}{n} \sum_{i=1}^{n} \left(x_i - \mathbf{D}^{(it-1)}\boldsymbol{\beta}_i\right)\boldsymbol{\beta}_i^T \tag{13}$$

Similar to the "dictionary update" stage, with the other variables fixed, we update the sparse coefficient **B** according to the following equation:

$$
\begin{aligned}
\min_{\mathbf{B}} \frac{1}{n} \sum_{i=1}^{n} \left\{ \left\| x_i - \mathbf{D}\boldsymbol{\beta}_i \right\|_2^2 + \lambda \left\| \boldsymbol{\beta}_i \right\|_1 \right\} + \mu \sum_{j=1}^{K} \delta \left\| \hat{\boldsymbol{\beta}}_j - \hat{v}_j \right\|_2^2 \\
= \min_{\mathbf{B}} \frac{1}{n} \sum_{i=1}^{n} \left\{ \left\| x_i - \mathbf{D}\boldsymbol{\beta}_i \right\|_2^2 + \lambda \left\| \boldsymbol{\beta}_i \right\|_1 + n\mu\delta \left\| \boldsymbol{\beta}_i - v_i \right\|_2^2 \right\}
\end{aligned}
\tag{14}
$$

For each *i*, Equation (14) is independent. Thus, finding the optimal $\{\boldsymbol{\beta}_i\}$ solves the following independent problems of the form:

$$
\min_{\boldsymbol{\beta}_i} \left\| x_i - \mathbf{D}\boldsymbol{\beta}_i \right\|_2^2 + \lambda \left\| \boldsymbol{\beta}_i \right\|_1 + n\mu\delta \left\| \boldsymbol{\beta}_i - v_i \right\|_2^2
\tag{15}
$$

Let $\Upsilon = \begin{bmatrix} x_i \\ \sqrt{n\mu\delta}v_i \end{bmatrix}$, $\Theta = \begin{bmatrix} \mathbf{D} \\ \sqrt{n\mu\delta}\mathbf{I} \end{bmatrix}$, and Equation (15) can be written as follows:

$$
\min_{\boldsymbol{\beta}_i} \left\| \Upsilon - \Theta\boldsymbol{\beta}_i \right\|_2^2 + \lambda \left\| \boldsymbol{\beta}_i \right\|_1
\tag{16}
$$

This is a combination of the quadratic and $\ell_1$ sparse terms; thus, the iterative shrinkage thresholding algorithm [46] is popular for solving this efficiently. In the *it*-th iteration, we obtain the following equation:

$$
\boldsymbol{\beta}_i^{(it)} = \begin{cases} \boldsymbol{v}^{(it-1)} - 2\lambda \cdot \mathrm{sgn}\left(\boldsymbol{v}^{(it-1)}\right) & \left| \Upsilon^{(it-1)} \right| > 2\lambda \\ 0 & \left| \Upsilon^{(it-1)} \right| \leq 2\lambda \end{cases}
\tag{17}
$$

where $\boldsymbol{v}^{(it-1)} = \frac{1}{2}\boldsymbol{\beta}_i^{(it-1)} + \left(\Theta^{(it-1)}\right)^T \left(\Upsilon^{(it-1)} - \Theta^{(it-1)}\boldsymbol{\beta}_i^{(it-1)}\right)$.

Finally, we update the variable **V** fixed **D** and **B**:

$$
\min_{\hat{v}_j \in \mathbf{V}} \sum_{j=1}^{K} \left[ \delta \left\| \hat{\boldsymbol{\beta}}_j - \hat{v}_j \right\|_2^2 + E\left(\hat{v}_j\right) \right],
\tag{18}
$$

which can be decomposed to *K* independent functions with respect to *j*:

$$
\min_{\hat{\mathbf{v}}_j} \delta \left\| \hat{\boldsymbol{\beta}}_j - \hat{\mathbf{v}}_j \right\|_2^2 + E\left(\hat{v}_j\right)
\tag{19}
$$

According to [44], we can obtain its solution as follows:

$$
\hat{v}_j = \begin{cases} \delta \left\| \hat{\boldsymbol{\beta}}_j \right\|_2^2 & \left\| \hat{\boldsymbol{\beta}}_j \right\|_2^2 < \frac{1}{\delta} \\ 1 & otherwise \end{cases}
\tag{20}
$$

---

**Algorithm 1: Adaptive Spectral Dictionary Learning**

---

**Input:** the training examples $\{\mathbf{x}_1, \dots, \mathbf{x}_n\}$, the regularization parameters $\lambda = 0.2$, $\mu = 0.001$.
**Initialize** $\delta = 1$, $it = 1$, $\mathbf{V}^0 = 0_{K \times n}$,
**while** $\delta < 10^6$ do
Input $\mathbf{V}^{it-1}$, $\mathbf{D}^{it-1}$, update $\mathbf{B}^{it}$ by (17);
Input $\mathbf{B}^{it}$, update $\mathbf{V}^{it}$ by (20);
Input $\mathbf{B}^{it}$ and $\mathbf{D}^{it-1}$, update $\mathbf{D}^{it}$ by (13);
$it \leftarrow it + 1$;
$\delta \leftarrow 2\delta$
end **while**
**Output:** the spectral dictionary $\mathbf{D} = \mathbf{D}^{it}$

---

### 2.2.2. Double $\ell_1$ Regularization

This subsection addresses the problem of how to obtain the sparse coefficient $\alpha_i$ from the HR MSI, **Y**, and the spectral dictionary, **D**, associated with Equation (5). Traditionally, $\alpha_i$ is obtained with a $\ell_1$ constraint as follows:

$$\alpha_i = \underset{\alpha_i}{\arg\min} \left\| y_i - \overline{\mathbf{D}}\alpha_i \right\|_2^2 + \lambda_1 \|\alpha_i\|_1 \tag{21}$$

where $\overline{\mathbf{D}} = \mathbf{PD}$. In Equation (21), the sparse coefficient of each $\alpha_i$ is computed independently. Actually, each pixel has a strong correlation with its nonlocal similar neighbors in the HR HSI. Thus, it is impossible for the sparse coefficients to be distributed randomly. In other words, even better SR results will be produced if the nonlocal similarities of the sparse coefficients are considered.

In Figure 3, we randomly plot the distributions of $\alpha_i - \sum_{j \in NN(i)} p_{ij}\alpha_j$ corresponding to the 23rd and 57th atoms (other atoms exhibit similar distributions) in the dictionary, **D**. Term $NN(i)$ is the index set of similar neighbors of $\alpha_i$. Moreover, $p_{ij} = \frac{1}{c} \exp\left(-\|\alpha_i - \alpha_j\|_2^2\right)$ are the weighting coefficients based on the similarity of $\alpha_i$ and $\alpha_j$, and $c$ is a positive constant. The number of nearest neighbors is selected as 10 in $NN(i)$. We can observe that the empirical distributions are highly peaked at zero and can be effectively characterized by $\ell_1$ functions, while $\ell_2$ functions have a much larger fitting error. Hence, this motivates us to improve the super-resolution quality by modelling the nonlocal similarity of the sparse coefficients by an $\ell_1$ prior.

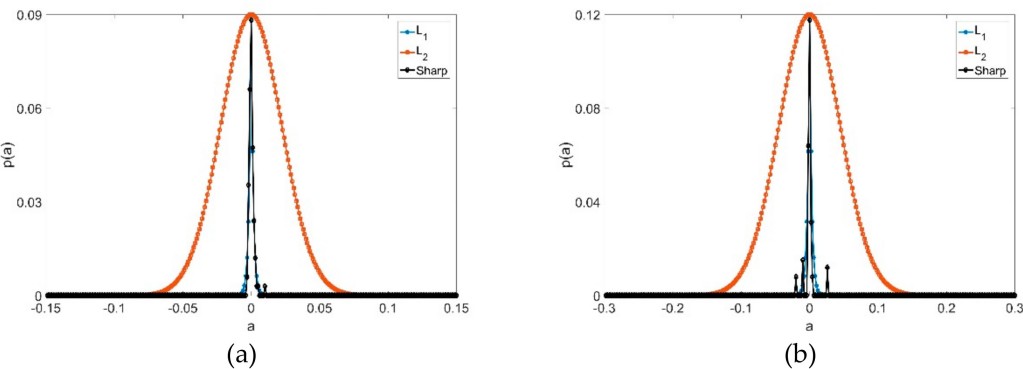

$$\qquad\qquad\qquad\qquad (a) \qquad\qquad\qquad\qquad\qquad\qquad\qquad\qquad (b)$$

**Figure 3.** The distribution of $\alpha_i - \sum_{j \in NN(i)} p_{ij}\alpha_j$ corresponding to the 23rd (**a**) and 57th (**b**) atom in the dictionary.

Based on this consideration, we incorporate the nonlocal geometric structure into the single $\ell_1$ constraint model as another regularization term for HSI SR as follows.

$$\alpha_i = \underset{\alpha_i}{\arg\min} \left\| y_i - \overline{\mathbf{D}}\alpha_i \right\|_2^2 + \lambda_1 \|\alpha_i\|_1 + \lambda_2 \left\| \alpha_i - \sum_{j \in NN(i)} p_{ij}\alpha_j \right\|_1 \tag{22}$$

where $\lambda_1$ and $\lambda_2$ are two regularization parameters. The second term is the sparse constraint on the coefficients, while the last term emphasizes the nonlocal similarity for the sparse coefficients.

Furthermore, $\alpha_i$ can be solved iteratively [47]. In the $l$-th iteration, we define $\kappa^{(l)} = \sum_{j \in NN(i)} p_{ij}\alpha_j^{(l)}$ and initialize $\kappa^{(0)} = 0$. Then we solve the single $\ell_1$ constraint model (21) to get $\alpha_i^{(1)} (i = 1, 2, \ldots, N)$. Based on $\left\{\alpha_i^{(1)}\right\}_{i=1}^N$, we can find similar sparse coefficients $\alpha_j^{(1)}$ ($j \in NN(i)$). In the next iteration of sparse coding process, $\alpha_i^{(2)}$ is the solution of Equation (22), and the second term is updated as

$\left\| \boldsymbol{\alpha}_i{}^{(2)} - \sum\limits_{j \in NN(i)} p_{ij}\boldsymbol{\alpha}_j{}^{(1)} \right\|_1$. Such a procedure is iterated until convergence. Thus, Equation (22) can be transformed as follows:

$$\boldsymbol{\alpha}_i{}^{(l)} = \underset{\boldsymbol{\alpha}_i}{\operatorname{argmin}} \left\| \mathbf{y}_i - \overline{\mathbf{D}}\boldsymbol{\alpha}_i{}^{(l)} \right\|_2^2 + \lambda_1 \left\| \boldsymbol{\alpha}_i{}^{(l)} \right\|_1 + \lambda_2 \left\| \boldsymbol{\alpha}_i{}^{(l)} - \sum\limits_{j \in NN(i)} p_{ij}\boldsymbol{\alpha}_j{}^{(l-1)} \right\|_1 \tag{23}$$

Regarding the *l*-th iteration, $\boldsymbol{\kappa}^{(l-1)} = \sum\limits_{j \in NN(i)} p_{ij}\boldsymbol{\alpha}_j{}^{(l-1)}$ is a constant. We simply rewrite Equation (23) as follows:

$$\boldsymbol{\alpha}_i{}^{(l)} = \underset{\boldsymbol{\alpha}_i}{\operatorname{argmin}} \left\| \mathbf{y}_i - \overline{\mathbf{D}}\boldsymbol{\alpha}_i{}^{(l)} \right\|_2^2 + \lambda_1 \left\| \boldsymbol{\alpha}_i{}^{(l)} \right\|_1 + \lambda_2 \left\| \boldsymbol{\alpha}_i{}^{(l)} - \boldsymbol{\kappa}^{(l-1)} \right\|_1 \tag{24}$$

Equation (24) is a double $\ell_1$ regularized least squares problem, which we solve by employing the surrogate functions [48]. Here, we introduce the following surrogate function:

$$\rho(\boldsymbol{\alpha}_i, a) = C\|\boldsymbol{\alpha}_i - a\|_2^2 - \left\| \overline{\mathbf{D}}\boldsymbol{\alpha}_i - \overline{\mathbf{D}}a \right\|_2^2 \tag{25}$$

where the constant, $C$, is chosen $\left( \left\| \overline{\mathbf{D}}^T\overline{\mathbf{D}} \right\|_2^2 < C \right)$ to make $\rho(\boldsymbol{\alpha}_i, a)$ convex, and $a$ denotes an auxiliary variable. Then we define the following function:

$$\begin{aligned} f\left(\boldsymbol{\alpha}_i{}^{(l)}, a\right) &= \left\| y_i - \overline{\mathbf{D}}\boldsymbol{\alpha}_i{}^{(l)} \right\|_2^2 + \lambda_1 \left\| \boldsymbol{\alpha}_i{}^{(l)} \right\|_1 + \lambda_2 \left\| \boldsymbol{\alpha}_i{}^{(l)} - \boldsymbol{\kappa}^{(l-1)} \right\|_1 \\ &\quad + C\left\| \boldsymbol{\alpha}_i{}^{(l)} - a \right\|_2^2 - \left\| \mathbf{D}\boldsymbol{\alpha}_i{}^{(l)} - \mathbf{D}a \right\|_2^2 \\ &= C\left\| \boldsymbol{\alpha}_i{}^{(l)} - \tau_i{}^{(l)} \right\|_2^2 + \lambda_1 \left\| \boldsymbol{\alpha}_i{}^{(l)} \right\|_1 + \lambda_2 \left\| \boldsymbol{\alpha}_i{}^{(l)} - \boldsymbol{\kappa}^{(l-1)} \right\|_1 + const \end{aligned} \tag{26}$$

where $\tau_i{}^{(l)} = \frac{1}{C}\left( \overline{\mathbf{D}}^T y_i - \overline{\mathbf{D}}^T\mathbf{D}a \right) + a$ and $const = \|y_i\|_2^2 + C\|a\|_2^2 - \left\| \overline{\mathbf{D}}a \right\|_2^2 - C\left\| \tau_i{}^{(l)} \right\|_2^2$. The objective function of (26) can be simplified further, as follows:

$$f\left(\boldsymbol{\alpha}_i{}^{(l)}\right) = \left\| \boldsymbol{\alpha}_i{}^{(l)} - \tau_i{}^{(l)} \right\|_2^2 + \mu_1 \left\| \boldsymbol{\alpha}_i{}^{(l)} \right\|_1 + \mu_2 \left\| \boldsymbol{\alpha}_i{}^{(l)} - \boldsymbol{\kappa}^{(l-1)} \right\|_1 \tag{27}$$

where $\mu_1 = \frac{\lambda_1}{C}$ and $\mu_2 = \frac{\lambda_2}{C}$ are regularization parameters. We can obtain the scalar version of the above minimization problem as follows:

$$g(m) = (m - m_1)^2 + \mu_1|m| + \mu_2|m - m_2| \tag{28}$$

where $m$, $m_1$, and $m_2$ are the scalar components of $\boldsymbol{\alpha}_i{}^{(l)}$, $\tau_i{}^{(l)}$, and $\boldsymbol{\kappa}^{(l-1)}$, respectively. Then, the solution to Equation (24) is given by the following equation:

$$\boldsymbol{\alpha}_i{}^{(l+1)} = \begin{cases} S_{\mu_1,\mu_2,\boldsymbol{\kappa}^{(l-1)}}\left(\tau_i{}^{(l)}\right) & \boldsymbol{\kappa}^{(l-1)} \geq 0 \\ -S_{\mu_1,\mu_2,-\boldsymbol{\kappa}^{(l-1)}}\left(-\tau_i{}^{(l)}\right) & \boldsymbol{\kappa}^{(l-1)} < 0 \end{cases} \tag{29}$$

The generalized shrinkage operator $S_{\mu_1,\mu_2,r_2}(r)$ is defined by the following:

$$S_{\mu_1,\mu_2,r_2}(r) = \begin{cases} r + \mu_1 + \mu_2 & r < -\mu_1 - \mu_2 \\ 0 & -\mu_1 - \mu_2 \leq r \leq \mu_1 - \mu_2 \\ r - \mu_1 + \mu_2 & \mu_1 - \mu_2 < r < \mu_1 - \mu_2 + r_2 \\ m_2 & \mu_1 - \mu_2 + r_2 \leq r \leq \mu_1 + \mu_2 + r_2 \\ r - \mu_1 - \mu_2 & \mu_1 + \mu_2 + r_2 < r \end{cases} \tag{30}$$

---

**Algorithm 2: Double $\ell_1$ Regularized Sparse Coding**

---

**Input:** the pixel set $\left\{y_1, \ldots, y_N\right\}$, the spectral dictionary (**D**), the transform matrix (**P**), the regularization parameters ($\lambda_1$ and $\lambda_2$), and the number of iterations ($L = 5$).
**For** $i = 1$ *to* $N$ do
Initialize $\alpha_i^{(0)} = 0$;
For $l = 1$ *to* $L$ do
$$\kappa^{(l-1)} = \sum_{j \in NN(i)} p_{ij}\alpha_j^{(l-1)}$$
$$\tau_i^{(l)} = \frac{1}{C}\left((\mathbf{PD})^T y_i - (\mathbf{PD})^T \mathbf{D}a\right) + a$$
$$\alpha_i^{(l+1)} = \begin{cases} S_{\mu_1,\mu_2,\kappa^{(l-1)}}\left(\tau_i^{(l)}\right) & \kappa^{(l-1)} \geq 0 \\ -S_{\mu_1,\mu_2,-\kappa^{(l-1)}}\left(-\tau_i^{(l)}\right) & \kappa^{(l-1)} < 0 \end{cases}$$
End For
End **For**
**Output**: the sparse coefficients $\mathbf{A} = \{\alpha_1, \ldots, \alpha_N\}$.

---

**Algorithm 3: HSI SR by Adaptive Dictionary Learning and Double $\ell_1$ Regularized Sparse Representation**

---

**Input:** LR HSI (**X**), HR MSI (**Y**), and the regularization parameters ($\lambda_1$ and $\lambda_2$).
(1) Learn the spectral dictionary, **D**, from **X** by using **Algorithm 1**;
(2) Obtain the sparse representation, **A**, from **Y** and **D** by using **Algorithm 2**.
**Output**: the HR HSI $\mathbf{Z} = \mathbf{DA}$.

---

## 3. Experimental Results and Analysis

In this section, we demonstrate the effectiveness of the proposed method on some popular datasets, using a series of experiments. Both qualitative and quantitative metrics are used to evaluate the performance.

### 3.1. Datasets and Experimental Setup

We performed verifying experiments on Cuprite and Pavia Center datasets, as shown in Figure 4. There are 105 spectral bands in the Cuprite and 102 bands in the Pavia Center dataset. We cropped each dataset to $480 \times 480$ pixels in spatial resolution. The real HSI of these two datasets were treated as ground-truth, and they were used to produce the simulated LR HSI and HR MSI. Specifically, the LR HSI, **X**, was generated by first applying a $9 \times 9$ Gaussian kernel of standard deviation 2 to the real HSI and then averaging pixels within an $s \times s$ window, where $s$ is the scaling factor (e.g., $s = 8, 16, 32$). For each dataset, we directly chose the blue, green, red, and near-infrared channels (corresponding to bands 7, 15, 25, and 42 in Cuprite, and bands 13, 33, 58, and 101 in Pavia Center, respectively) of ground-truth, to simulate the HR MSI, **Y**. To facilitate the numerical calculation, the intensities of each band in HSI were normalized to [0, 255].

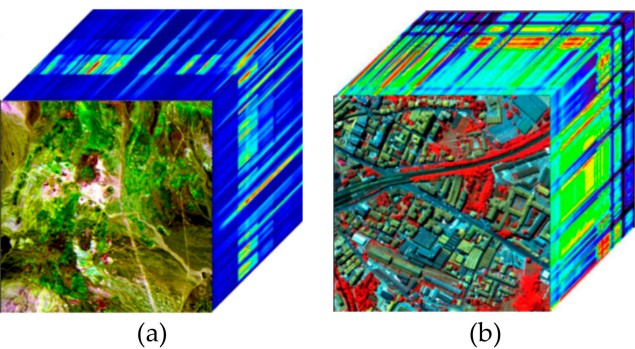

(a)                 (b)

**Figure 4.** The data cubes: (**a**) Cuprite and (**b**) Pavia Center.

The proposed method is compared with five representative algorithms: SASFM [32], G-SOMP+ [34], SSR [40], NNSR [38], and SSCSR [39]. To ensure the reliability of the results, we repeat each super resolution method 20 times on the test datasets.

### 3.2. Quality Metrics

We adopt three quantitative measures for the evaluation: relative dimensionless global error in synthesis (ERGAS) [49], root mean square error (RMSE), and spectral angle mapper (SAM) [50].

The RMSE measures the deviation between the reference HR HSI, **R**, and the reconstructed HR HSI, **Z**:

$$\text{RMSE} = \sqrt{\frac{\|\mathbf{R} - \mathbf{Z}\|_F^2}{BN}} \tag{31}$$

The ERGAS metric [49] calculates the average amount of spectral deviation in each band, as defined below:

$$\text{ERGAS} = \frac{100}{s} \cdot \sqrt{\frac{1}{B} \sum \frac{\text{RMSE}(r_b, z_b)}{\mu_{r_b}}} \tag{32}$$

where $s$ is the scaling factor, $r_b$ and $z_b$ represents the $b$-th band of **R** and **Z**, respectively, and $\mu_{r_b}$ is the mean of $r_b$.

At last, we calculated SAM [50], which is defined as the angle between two spectral vectors, $r_b$ and $z_b$, averaged over all pixels:

$$\text{SAM} = \frac{1}{N} \sum \arccos\left(\frac{r_b{}^T z_b}{(r_b{}^T r_b)^{1/2} \cdot (z_b{}^T z_b)^{1/2}}\right) \tag{33}$$

According to the above definition, the smaller the RMSE, ERGAS, and SAM metrics, the better the super-resolution performance.

### 3.3. Performance Comparison of Different Methods

Table 1 shows the average RMSE, ERGAS, and SAM results of the two datasets under different downsampling factors by the six compared methods. Our approach outperforms the others in terms of the RMSE, ERGAS, and SAM results, which clearly indicates that the adaptive learned dictionary can exploit the underlying structures in the HSI. The double $\ell_1$ regularized sparse representation illustrates the superior performance over other competing methods. Thus, these numerical results validate the power of the proposed model for HSI super-resolution.

To facilitate visual comparison, Figure 5c–h shows the reconstructed HR HSI of band 100 by different competing approaches, with a scaling factor of $s = 8$ on the Cuprite dataset. We can see that all compared methods can generate spatial structures very well. To describe the differences of different methods more intuitively, Figure 5i–n presents a comparison of the differences (absolute value) in pixel values between each reconstructed image and the reference HR image. The mountain and river regions are not well preserved, resulting in large errors in the SASFM. The NNSR and SSCSR methods deliver a better result, but still cannot recover the missing details in the mountain. By contrast, our method can recover more details. Thus, the smallest reconstruction errors are achieved in Figure 5n.

To illustrate the consistency of the overall performance, we present the related results of the same band (band 100) with another scaling factor ($s = 32$) on the Cuprite dataset in Figure 6. We can draw the same conclusion that the six compared methods can significantly enhance the spatial resolution of the input LR HSI. However, it should be noted that our method achieves the best result.

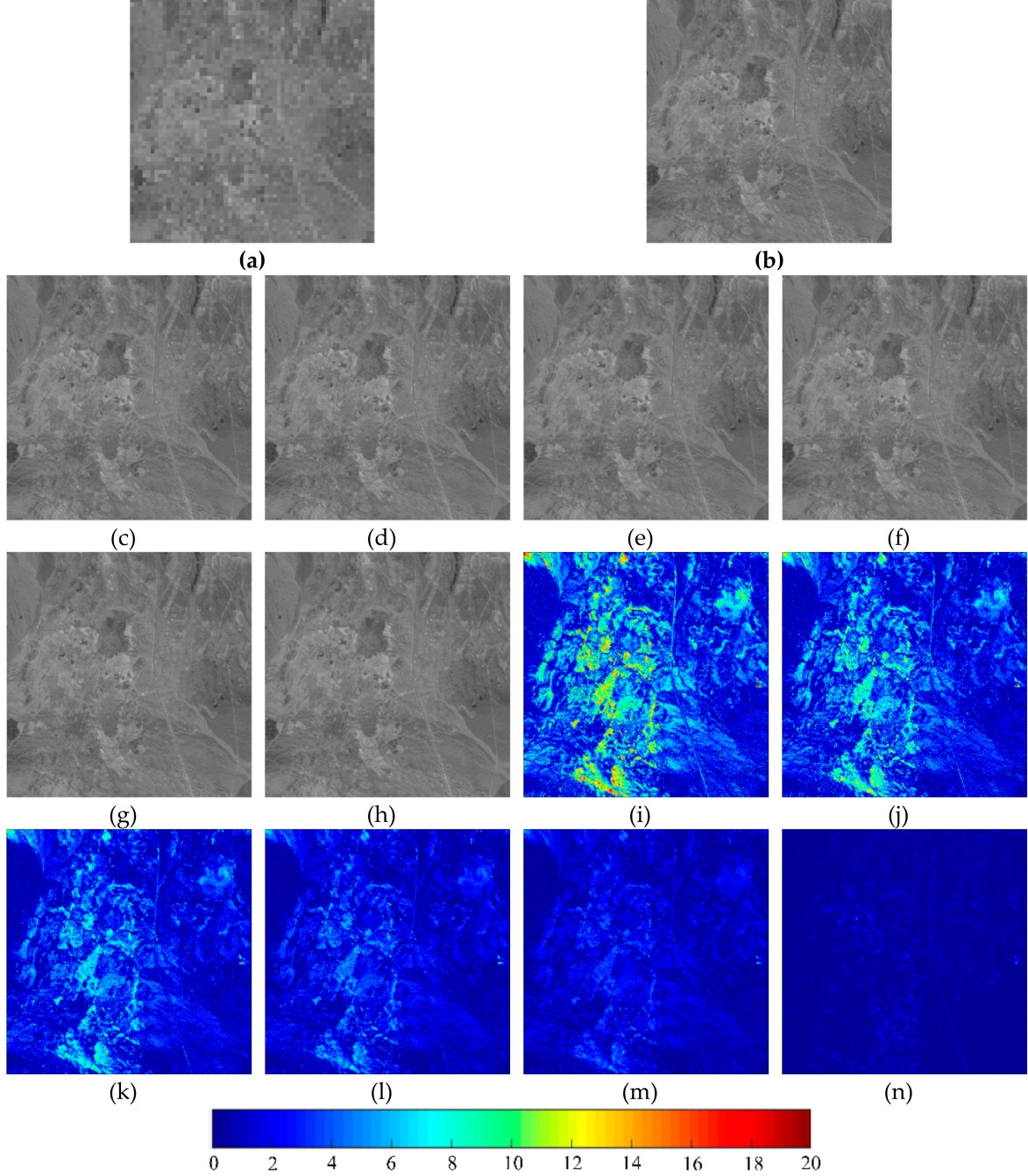

**Figure 5.** SR results of the 100th band of the Cuprite dataset with a scaling factor of *s*= 8. The second row presents the reconstructed images achieved by the different methods. The third row shows the corresponding error images. (**a**) LR image, (**b**) original HR HSI image, and SR results by different methods: (**c**) SASFM, (**d**) G-SOMP+, (**e**) SSR, (**f**) NNSR, (**g**) SSCSR, (**h**) Proposed; corresponding error images: (**i**) SASFM, (**j**) G-SOMP+, (**k**) SSR, (**l**) NNSR, (**m**) SSCSR, (**n**) Proposed.

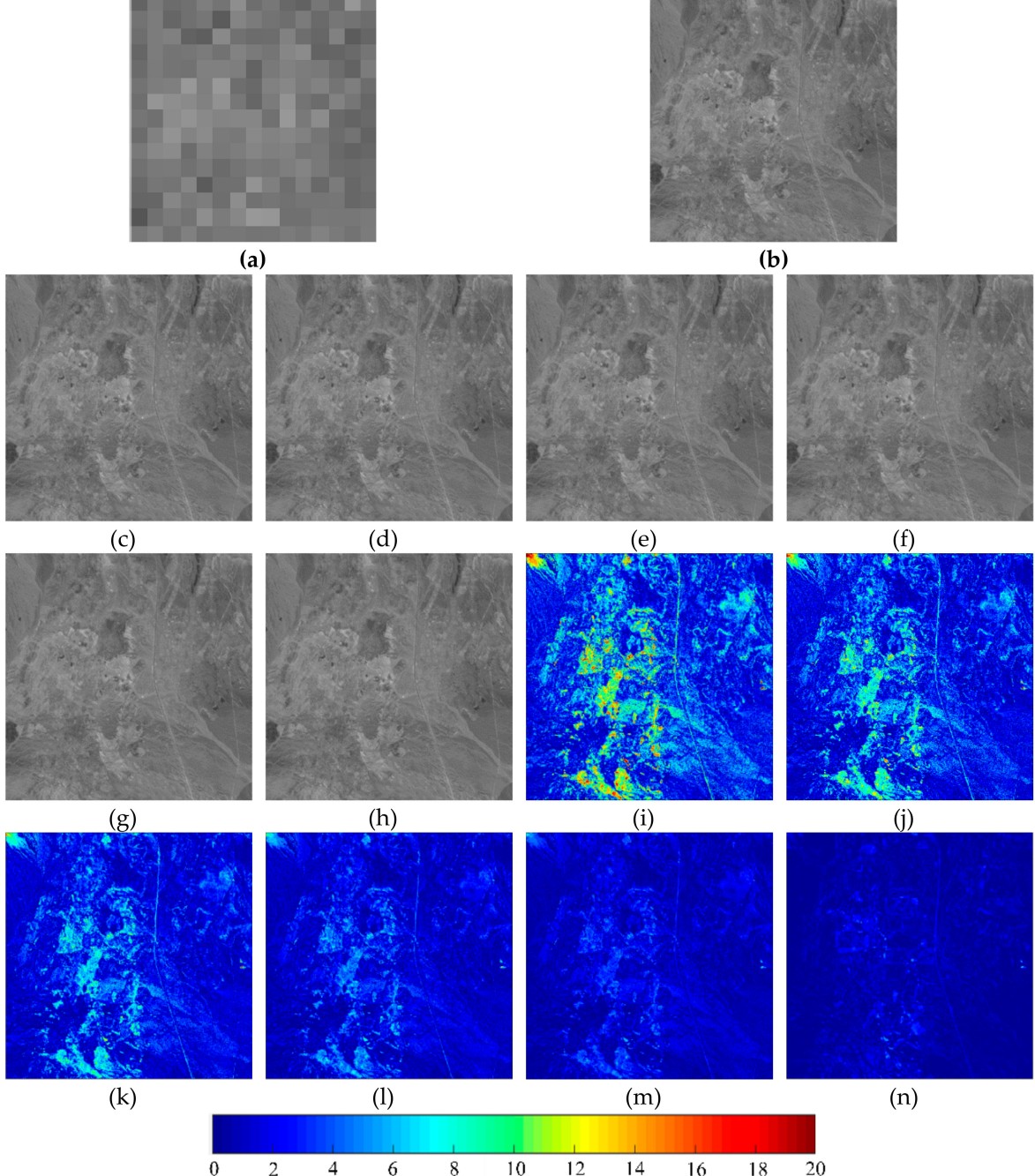

**Figure 6.** SR results of the 100th band of the Cuprite dataset with a scaling factor of *s*= 32. The second row presents the reconstructed images achieved by the different methods. The third row shows the corresponding error images. (**a**) LR image, (**b**) original HR HSI image, and SR results by different methods: (**c**) SASFM, (**d**) G-SOMP+, (**e**) SSR, (**f**) NNSR, (**g**) SSCSR, (**h**) Proposed; corresponding error images: (**i**) SASFM, (**j**) G-SOMP+, (**k**) SSR, (**l**) NNSR, (**m**) SSCSR, (**n**) Proposed.

**Table 1.** Quantitative measures by different methods on Cuprite and Pavia Center.

| Downsampling Factor | Methods | Cuprite | | | Pavia Center | | |
|---|---|---|---|---|---|---|---|
| | | RMSE | ERGAS | SAM | RMSE | ERGAS | SAM |
| $s = 8$ | SASFM | 1.0065 | 0.5987 | 2.1624 | 1.8734 | 0.7003 | 2.3561 |
| | G-SOMP+ | 0. 8410 | 0.5683 | 1.8999 | 1.5552 | 0.5826 | 1.9708 |
| | SSR | 0.7627 | 0.5511 | 1.8239 | 1.2390 | 0.5747 | 1.9089 |
| | NNSR | 0.6373 | 0.4562 | 1.6120 | 1.1537 | 0.5521 | 1.8295 |
| | SSCSR | 0.5663 | 0.3318 | 1.2278 | 0.0990 | 0.5159 | 1.7566 |
| | Proposed | *0.4852* | *0.2961* | *0.9088* | *1.0574* | *0.4975* | *1.6388* |
| $s = 16$ | SASFM | 1.1818 | 0.3136 | 1.9852 | 2.1621 | 0.3519 | 2.4395 |
| | G-SOMP+ | 0.9109 | 0.2796 | 1.7674 | 1.7861 | 0.3009 | 2.1477 |
| | SSR | 0.8567 | 0.2788 | 1.7585 | 1.3697 | 0.2978 | 2.1344 |
| | NNSR | 0.7629 | 0.2005 | 1.5662 | 1.2790 | 0.2977 | 1.9591 |
| | SSCSR | 0.6937 | 0.1811 | 1.3039 | 1.2190 | 0.2891 | 1.8889 |
| | Proposed | *0.5474* | *0.1695* | *1.0157* | *1.1904* | *0.2837* | *1.8650* |
| $s = 32$ | SASFM | 1.4076 | 0.1629 | 2.0011 | 4.2323 | 0.3002 | 4.0387 |
| | G-SOMP+ | 1.1267 | 0.1436 | 1.8117 | 3.8084 | 0.2511 | 3.3054 |
| | SSR | 0.9845 | 0.1397 | 1.7546 | 3.4486 | 0.2332 | 2.9203 |
| | NNSR | 0.8393 | 0.1128 | 1.4658 | 2.8099 | 0.2008 | 2.5580 |
| | SSCSR | 0.7654 | 0.1095 | 1.3267 | 2.2790 | 1.1744 | 2.3016 |
| | Proposed | *0.6826* | *0.1015* | *1.1990* | *2.1847* | *0.1575* | *2.0664* |

To verify the robustness of the proposed method, Figures 7 and 8 present the SR results of different methods on the Pavia Center dataset, which has more varied content in the captured area. When the scaling factor ($s = 8$) is small, the building regions are not well reconstructed by the SASFM method, as shown in Figure 7i. When we increase the scaling factor ($s = 32$), larger errors occur in the SASFM reconstructed image (the edges of the buildings and the river are clearly visible in Figure 8i). This is because SASFM ignored the structural similarity in the MSI during the SR process. Although the GSOMP+ method made use of the structure similarity, it was only exploited with a fixed window; hence, it cannot use the spatial information sufficiently. Accordingly, the outlines of the buildings and the river are still visible in Figures 7j and 8j. Figures 7l and 8l demonstrate that the nonnegative structured sparse-representation model can efficiently preserve many details and edges. From Figure 7m,n and Figure 8m,n, we can observe that the proposed method slightly outperforms the SSCSR method [30] in recovering the details of the original HSI.

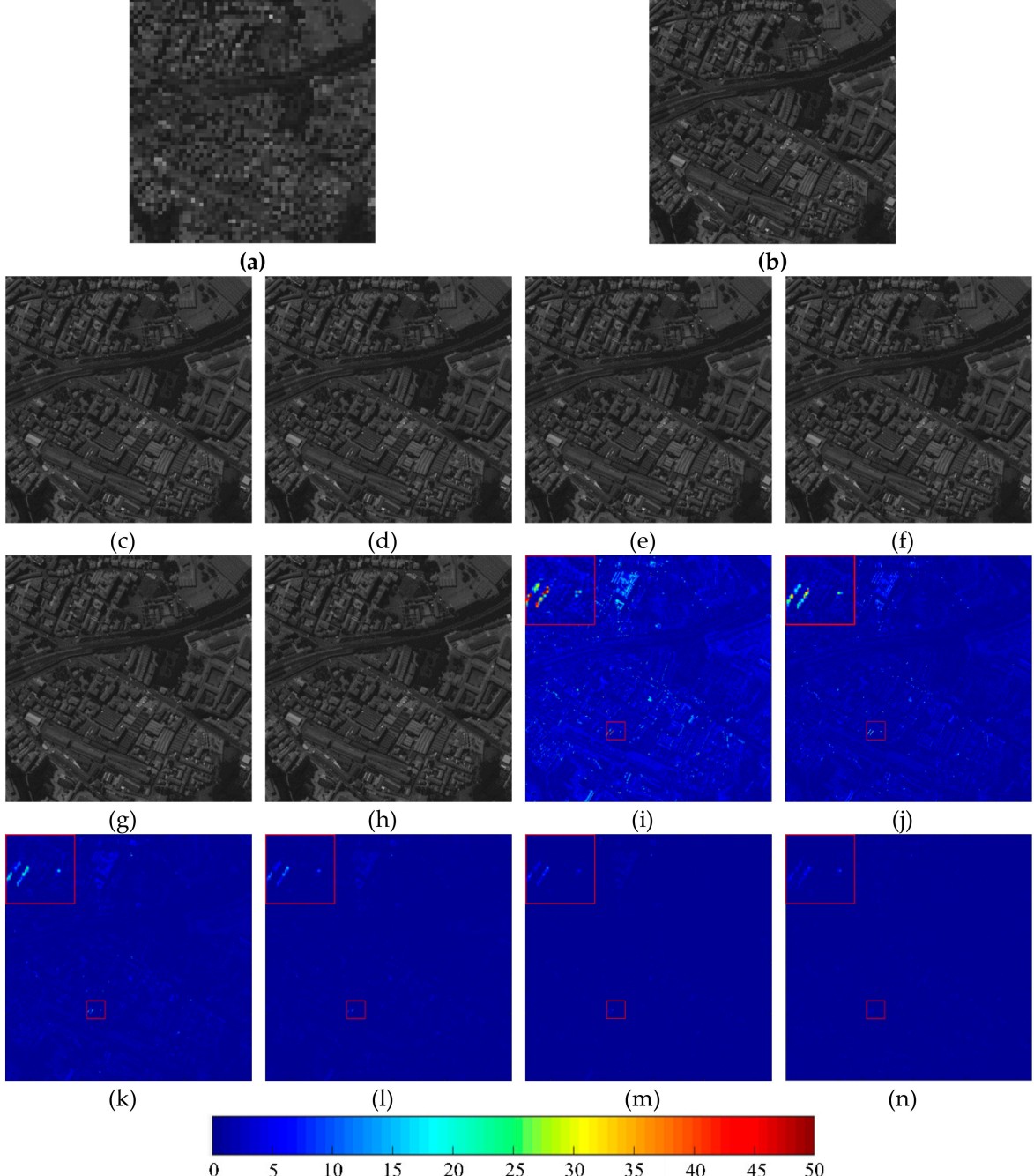

**Figure 7.** SR results of the 48th band of the Pavia Center dataset with a scaling factor of $s = 8$. The second row presents the reconstructed images achieved by the different methods. The third row shows the corresponding error images. (**a**) LR image, (**b**) original HR HSI image, SR results by different methods: (**c**) SASFM, (**d**) G-SOMP+, (**e**) SSR, (**f**) NNSR, (**g**) SSCSR, (**h**) Proposed; corresponding error images: (**i**) SASFM, (**j**) G-SOMP+, (**k**) SSR, (**l**) NNSR, (**m**) SSCSR, (**n**) Proposed.

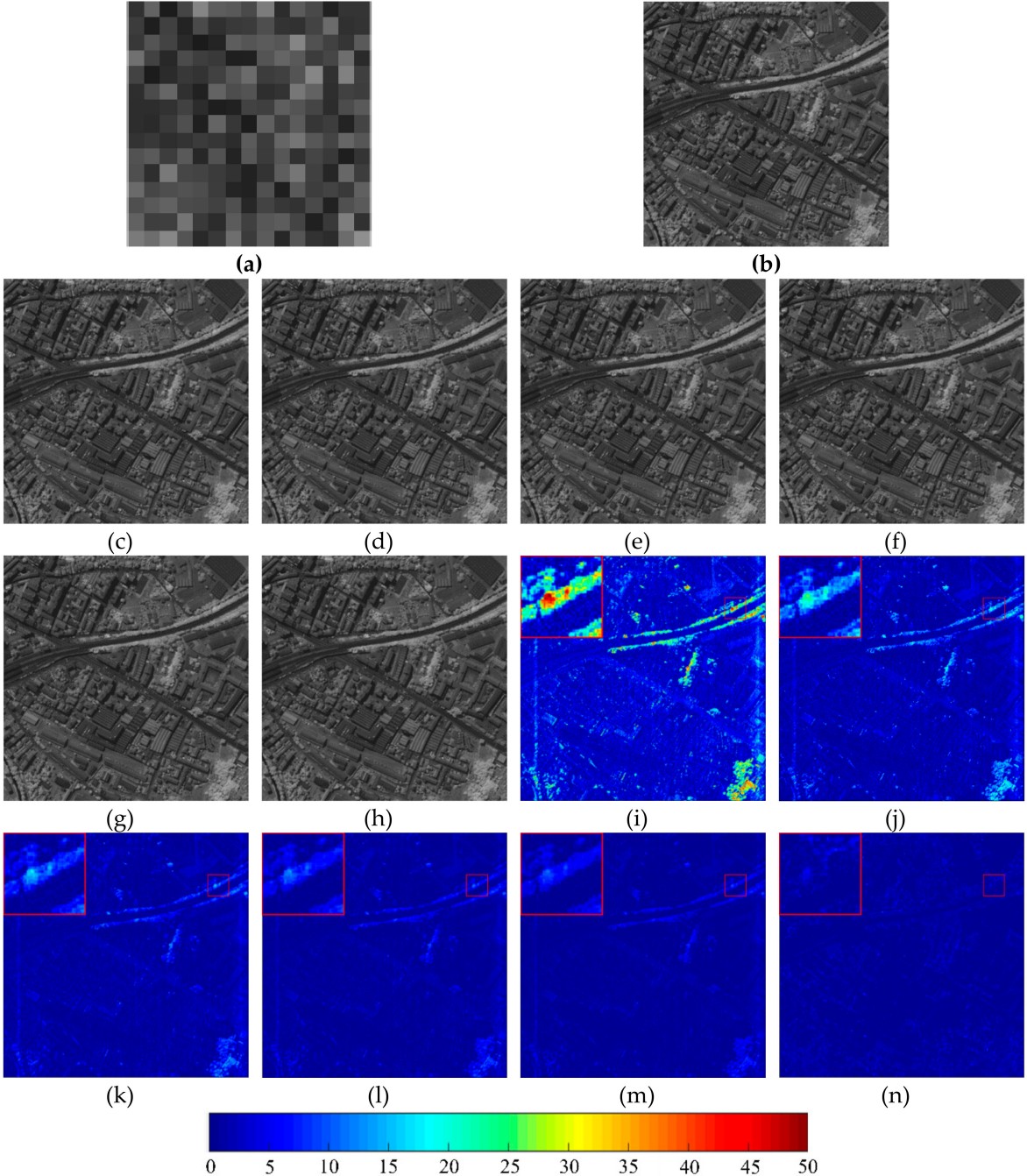

**Figure 8.** SR results of the 94th band of the Pavia Center dataset with scaling factor of $s= 32$. The second row presents the reconstructed images achieved by the different methods. The third row shows the corresponding error images. (**a**) LR image, (**b**) original HR HSI image, SR results by different methods: (**c**) SASFM, (**d**) G-SOMP+, (**e**) SSR, (**f**) NNSR, (**g**) SSCSR, (**h**) Proposed; corresponding error images: (**i**) SASFM, (**j**) G-SOMP+, (**k**) SSR, (**l**) NNSR, (**m**) SSCSR, (**n**) Proposed.

### 3.4. Effects of the Adaptive Size Dictionary

(1) In Figure 9, we present the values of objective function (10) vs. the iteration times. This proves that the proposed algorithm terminates in finite steps. The optimal value of the objective function is achieved after 15 iterations for each dataset. In Algorithm 1, a random dictionary is selected for the initialization. Thus, further experiments are conducted to evaluate the sensitivity of our learning algorithm to different starting points. For each dataset, we generate 200 different initial dictionaries

randomly, starting from which we can train different dictionaries. The related statistics on the final learned dictionary sizes are listed in Table 2. The variance of the sizes is very small. The minimum size is very similar to the maximum size. This suggests that different initial dictionaries have little effect on the final training results.

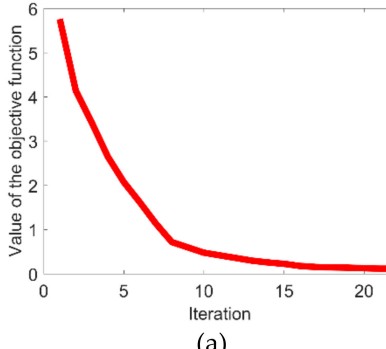
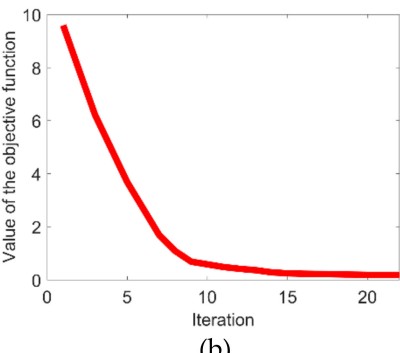

(a)                                    (b)

**Figure 9.** Value of the objective function evaluated by the proposed learning method with a scaling factor of 32 on different datasets: (**a**) Cuprite and (**b**) Pavia Center.

**Table 2.** Average sparse reconstruction errors by different learning methods on Cuprite and Pavia Center.

| Downsampling Factor | Cuprite | | Pavia Center | |
|---|---|---|---|---|
| | **Traditional** | **Adaptive** | **Traditional** | **Adaptive** |
| $s = 8$ | 11.1233 | 2.1488 | 19.0409 | 6.3434 |
| $s = 16$ | 2.1488 | 1.9752 | 15.5154 | 2.9938 |
| $s = 32$ | 0.6115 | 0.5165 | 10.3444 | 0.7820 |

(2) To evaluate the adaption of our learning strategy, the traditional dictionary learning method (i.e., Equation (7)) and our proposed method are applied to the test datasets (see Figure 4). The dictionary size is set to 300, according to Equation (7), while 38 and 43 reflectance vectors are sufficient to represent the structure variations by our adaptive strategy for the Cuprite and Pavia Center datasets, respectively. The experimental results indicate that the sizes of the learned dictionaries vary significantly according to different learning methods. This demonstrates that our estimated dictionary sizes are adaptive and suitable. For quantitative evaluation, the average sparse reconstruction error, $\frac{1}{n}\sum_{i=1}^{n}\left\|x_i - \mathbf{D}\boldsymbol{\beta}_i\right\|_2^2$, is calculated for each test dataset in Table 2. We can see that our learned dictionaries can reduce the redundancy effectively and avoid the interference of nonspecific errors. This demonstrates the efficiency of the proposed optimization algorithms.

(3) As we know, K-SVD [51] is popular for learning fixed-size dictionaries. Thus, we compare the proposed adaptive dictionary learning method with the K-SVD method. To ensure the reliability of the K-SVD algorithm, we executed 50 iterations to train the spectral dictionary by using orthogonal matching pursuit (OMP) [52] for sparse coding with $T = 6$ ($T$ is the sparsity of each trained atom). Finally, the number of atoms for K-SVD is chosen as 300, which is the best or nearly the best atom number according to 20 experiments. Although the resultant images produced by K-SVD have good visual quality, K-SVD is less able to sharpen the details in the mountain area. Fortunately, the adaptive dictionary can effectively reconstruct most spatial details with less obvious spectral distortions, as shown in Figure 10e,f. In other words, it is not effective to reflect the complex structures of HSI by a dictionary with a fixed size. Their numerical results from different test datasets are reported in Table 3.

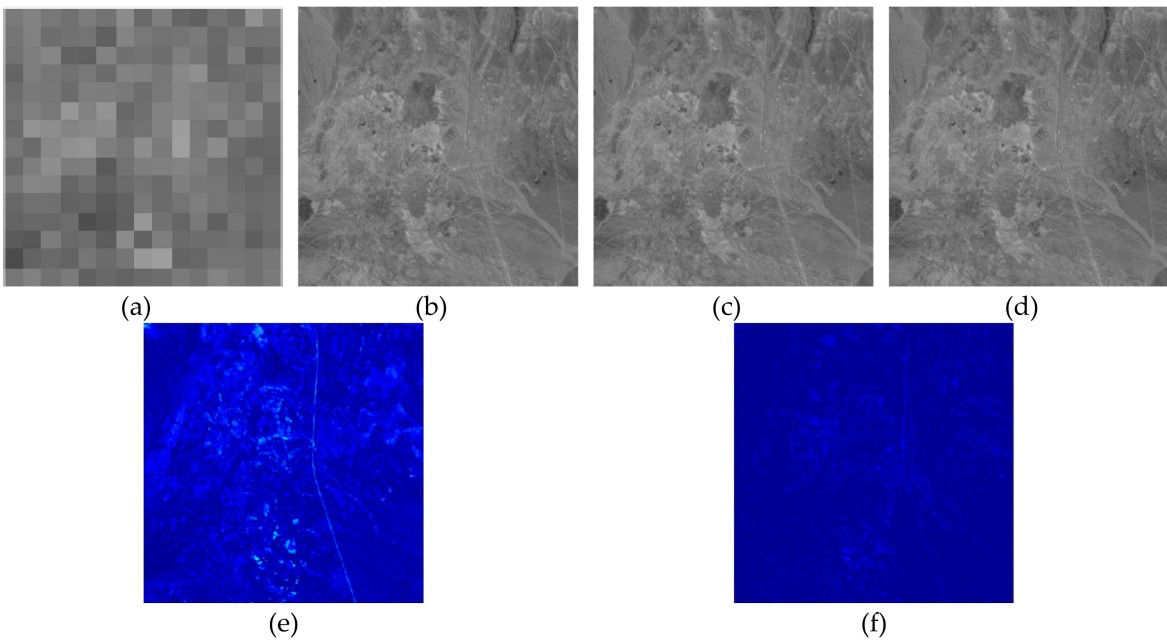

**Figure 10.** Images of the 70th band of the Cuprite dataset. (**a**) LR image; (**b**) original HR image; (**c**) and (**d**) are the SR results by K-SVD and adaptive dictionary learning method, respectively; (**e**) and (**f**) are the corresponding error images.

**Table 3.** Super-resolution results by K-SVD and adaptive dictionary learning on the Cuprite (the top values) and Pavia Center (the bottom values) datasets.

| Method | RMSE | ERGAS | SAM |
|:---:|:---:|:---:|:---:|
| **K-SVD** | 1.0451 | 0.1322 | 1.6000 |
| | 3.2085 | 0.2288 | 3.0525 |
| **Adaptive dictionary** | 0.6826 | 0.1015 | 1.1900 |
| | 2.18747 | 0.1575 | 2.0664 |

### 3.5. Parameters Analysis

The regularization parameters, $\lambda_1$ and $\lambda_2$, balance the different contributions of regularization terms and need to be carefully tuned in Equation (21). Thus, many experiments are performed to demonstrate the effectiveness of the proposed regularizations. Figures 11 and 12 present the curves of RMSE variations with different parameters and scaling factors ($s = 8, 16, 32$) on the Cuprite and Pavia Center datasets, respectively. With a smaller value of $\lambda_1$ (between 0.005 and 0.01), the proposed method generates unsatisfactory results. As $\lambda_1$ increases, better results can be achieved, as shown by the top rows in Figures 11 and 12. However, we also note that the performance exhibits a descending trend with increasing values of $\lambda_1$ (larger than 0.02). Nonetheless, when the values of $\lambda_1$ are between 0.01 and 0.02, the RMSE measurements are stable. According to the bottom rows in Figures 11 and 12, we can see that the optimal value of $\lambda_2$ is $8 \times 10^{-5}$. Moreover, when the values depart significantly from $8 \times 10^{-5}$, the corresponding RMSE values clearly fluctuate. The above experimental results demonstrate that our proposed combination can achieve promising results. In view of this, to achieve optimal or nearly optimal performance, we recommend setting $\lambda_1 \in [0.01, 0.02]$ and $\lambda_2 = 8 \times 10^{-5}$.

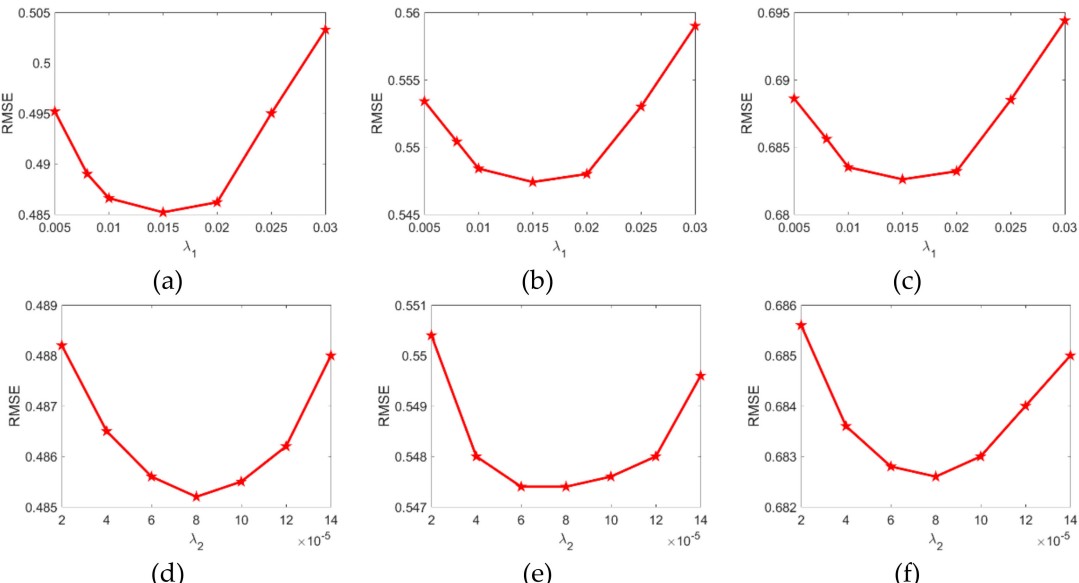

**Figure 11.** RMSE curves of different parameters and scaling factors on the Cuprite dataset. The top and bottom rows show the RMSE results with different values of $\lambda_1$ and $\lambda_2$, respectively. From left to right, RMSE variations with different scaling factors: (**a**,**d**) $s = 8$; (**b**,**e**) $s = 16$; and (**c**,**f**) $s = 32$.

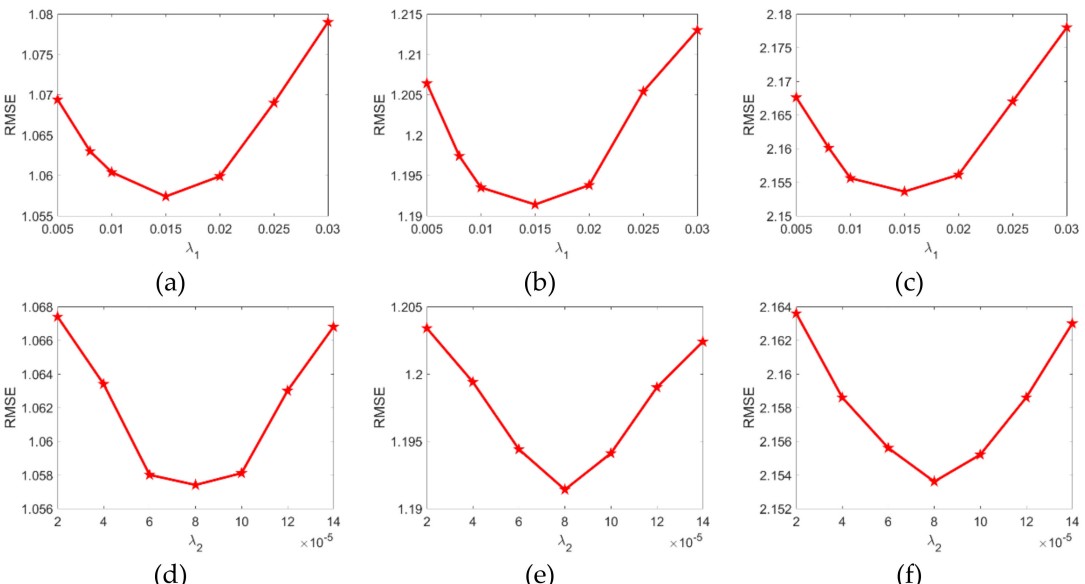

**Figure 12.** RMSE curves of different parameters and scaling factors on the Pavia Center dataset. The top and bottom rows show the RMSE results with different values of $\lambda_1$ and $\lambda_2$, respectively. From left to right, RMSE variations with different scaling factors: (**a**,**d**) $s = 8$; (**b**,**e**) $s = 16$; and (**c**,**f**) $s = 32$.

*3.6. Discussion on Computational Complexity*

The proposed SR method incurs major costs from two aspects: the spectral dictionary learning and the sparse representation. For each step in Algorithm 1, the computation of the spectral dictionary learning is $O(K^2n + 2KnB)$. For the parameter $\delta$, it needs $log_2\delta$ steps to achieve the optimal result. The overall complexity of Algorithm 1 is $O(log_2\delta(K^2n + 2KnB))$. In Algorithm 2, the complexity of updating the sparse coefficients is $O(LK)$ for each pixel; therefore, $N$ pixels need approximately $O(NLK)$.

The CPU times of different SR methods are presented in Table 4. All the algorithms were implemented with MATLAB on an Intel Core i7-6820 2.7-GHz CPU. The SASFM method has the fastest

running time through the sparse coding technique of OMP [52]. The SSR and SSCSR methods have a long processing time due to the clustering-based sparse representation framework. Our proposed method runs quite slowly and requires approximately 5 min. In the future, we can expect to speed up the proposed algorithm by using a graphics-processing unit.

**Table 4.** Average running time (seconds) of the compared methods on the simulated datasets with a scaling factor of 32.

| Method | SASFM | G-SOMP+ | SSR | NNSR | SSCSR | Proposed |
|--------|-------|---------|-----|------|-------|----------|
| Time   | 11    | 235     | 579 | 146  | 723   | 276      |

## 4. Discussion

Compared to other methods, the proposed method achieves a superior SR performance. There are mainly two reasons why. First, the adaptive dictionary can represent different variations reasonably compared with traditional dictionary learning methods. Second, the nonlocal similarities of the sparse coefficients are exploited to improve the HSI SR quality.

From the parameter analysis, we can find that the RMSE stay relatively stable, without any incremental performance, when the parameters are set to the recommended values (see Figures 11 and 12). In other words, the performance of our proposed method is robust. From the comparison of the execution time, we observe that the proposed model is not computationally efficient (see Table 4). However, our method could achieve better performance in comparison with other methods on the two hyperspectral datasets. In light of this, it will be interesting to design an architecture with multicore CPU [53], to optimize the execution time.

## 5. Conclusions

This paper proposed a new and effective method for HSI super-resolution based on sparse representation. There are two distinctive features of the proposed method. On the one hand, an adaptive learning strategy is used to learn a spectral dictionary, which represents different content and features reasonably. On the other hand, double $\ell_1$ regularized constraints are employed to characterize the similarities of the sparse coefficients, to improve the HSI SR quality. Extensive experimental results from two popular HSI datasets validate the superior performance of the proposed method over other competitive methods. The experiments of the parameters demonstrate the robustness of the proposed method.

In the future, we can combine the tensor model [54] and the shape-adaptive technique [55,56] to explore the spatial–spectral information adaptively and sufficiently. Deep-learning approaches have recently gained great attention in many fields [57–64]. It will be a new task to design a deep architecture to improve the performance of the HSI SR.

**Author Contributions:** Experiments and writing, S.T.; supervision, Y.X.; review and editing, L.S. and L.H.

**Funding:** This research was supported in part by the Fundamental Research Funds for the Central Universities (Grant No. LGZD201702, LGYB201807), the Natural Science Foundation of Jiangsu Province (Grant No. BK20171074 and BK 20150792), and the National Natural Science Foundation of China (Grant No. 61702269 and 61971233).

**Acknowledgments:** The authors would like to thank the Assistant Editor who handled our paper and the anonymous reviewers for providing help comments that significantly helped us improve the technical quality and presentation of our paper.

**Conflicts of Interest:** The authors declare no conflicts of interest.

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
