# Peer review of "Hyperspectral Image Super-Resolution via Adaptive Dictionary Learning and Double"

_remotesensing, doi:10.3390/rs11232809_

Round 1
Reviewer 1 Report
Dear Authors I have review the manuscript, Hyperspectral Image Super-resolution via Adaptive Dictionary Learning and Double Constraint.
the email address of the first author and the corresponding does not match.
The method proposed could have a feasibility in real applications. The paper is well written and the rationale of the experiment is clear for the reader.
I cannot fully assess the math so I would prefer that the Editor or a specific mathematician could assess their correctness.
The idea is ambitious, the paper is written in a good English.
Then, limitations due to ….
Question that could be address from a environmental point of view are: are the methods good for natural as well as anthropic environment equally?
What about the applicability of the method at large scale?
Though I trust the authors I cannot assess the plagiarism (if any) so I ask the journal to do this,
I wish you good luck
Kind regards
Detailed comments:
Line 54:missing reference
Author Response
Thank you for your comments. Please find the response to reviewers in the attachment.

Reviewer 2 Report
General comments
This manuscript by Tang and Xu proposed a new technique for enhancing the resolution of hyperspectral images based on sparse representation. The main objectives of the paper are clear, and the abstract is concise and straightforward. Generally, the manuscript is well-written and has a well-organized sequence. However, I have some suggestions for the authors to consider to improve the quality of the manuscript as follows:
1 - I suggest including a materials and methods (methodology) section just right after the introduction section. Furthermore, it would be good if the authors emphasize on the main aim of the paper in the last paragraph of the introduction.
2 - The discussion section needs to be extended to interpret the manuscript’s results in a better fashion.
3 - The authors need to relate the conclusions to their best quantitative results and findings. As it stands, the conclusions section is missing crucial details.
Author Response

(The authors gave the same response as above.)

Reviewer 3 Report
Thanks for this article about increasing resolution of hyperspectral image with adaptive dictionary.
Summary
The article starts with explanation on hardware constraints what impact a choice between high resolution and high spectral definition. The community use to combine low resolution hyper-spectral image with high resolution multi-spectral image.
The article simply and clearly presents the mathematical bases of that known process and point the terms involved to improved the quality of the results.
Then the main concepts are explain: adaptive dictionary and Double l1 regularization.
Some algorithms support the explanations.
The article described the datasets used (Cuprite and Pavia Center).
It explains the evaluation and gives visual results (figures) and numeric results (tables).
The article is supported by many recent references.
56 references (45% after 2014, 25% before 2006)
Strengths
Simplicity and efficiency of method explanations.
Weakness
Algorithms description should be improved
Comments
L201: A confusion is made between assignment and equality test.
use “i ← i+1” instead “i = i+1”. Remove all “=” in the algorithm.
What “Fix” does mean? Does it mean “input parameter”?
“D = D0” before while is never used (remove it).
The result is Dit
The computation of equations (12), (16), and (19) must be develop as algorithms or function calls (clearly described).
L253: same comment as L201.
remove all “=” signs.
develop equation (29) to show the loop on Ai.
Showing all loops in an algorithm gives visually the complexity.
L217: caption must be attached to the same page with the figure.
L278: The evaluation is not enough details. How we can trust the quantitative measures? What did they compute exactly?
L410: In the table2 the line below “factor” is smaller than all others.
L484: The conclusion should also indicates future works (L478).
L516, L518, L552, L560, L565, L569, L578, L581, L587, L597, L600, L604, L621, L624, L631: dates not bold
Author Response

(The authors gave the same response as above.)

Reviewer 4 Report
I truly appreciated reading your manuscript. It is generally clearly written, the literature review is thorough and well structured and the numerical experiments are detailed and well documented. In my opinion your work could be a nice contribution to the field, provided that some minor issues are faced. In the following I report my suggestions.
1) The formal correctness of the methodology description (Section 2) should be improved:
- some of the matrices and vectors dimensions are missing (H,T,D, alpha_i, A)
- Eq(4) should be rewritten (maybe in matrix notation?) to explain the h_i terms (are they row/columns of H?) and some terms are not even defined (t_i, beta_i). Moreover, can you check (according to their definition, now missing) that the dimensions of h_i and alpha_i match?
- line 129: vector y_i, according to (2) and (3), should include an error term. What you write in (5) is an estimate, I suppose
2) It seems to me that Section 3.2 is a (shortened) reformulation of your previous work, as published in ref [43], so that some points are quite unclear. You should probably cite that work and refer to it for further details.
Finally, I report some points to be corrected:
- reference [36] in line 191 is wrong. Probably it should be [45]. And please, check that all the listed literature is actually cited in the manuscript.
- the constant c (line 209) is not defined;
- line 223: it should be “regularization parameters”, not “regularized parameters”
- line 225: it should be “solve for alpha_i”
- the iterative procedure (line 228) is unclear;
- the mathematical expressions (22),(23),(24), (26) and (27) are NOT Equations
(please correct at lines 229,232,234,238,241,247).
- reference [39] in line 235 is wrong. Probably it should be [48].
- line 238: “Then Equation (24) becomes” should probably read “Then from (24) we can define”
Author Response

(The authors gave the same response as above.)

Reviewer 5 Report
I have no comments. I like the article. I have no I . I like the article. I have no comments. I like the article.Author Response
Thank you for your hard wrok.